# Potential Antiviral Options against SARS-CoV-2 Infection

**DOI:** 10.3390/v12060642

**Published:** 2020-06-13

**Authors:** Aleksandr Ianevski, Rouan Yao, Mona Høysæter Fenstad, Svetlana Biza, Eva Zusinaite, Tuuli Reisberg, Hilde Lysvand, Kirsti Løseth, Veslemøy Malm Landsem, Janne Fossum Malmring, Valentyn Oksenych, Sten Even Erlandsen, Per Arne Aas, Lars Hagen, Caroline H. Pettersen, Tanel Tenson, Jan Egil Afset, Svein Arne Nordbø, Magnar Bjørås, Denis E. Kainov

**Affiliations:** 1Department of Clinical and Molecular Medicine, Norwegian University of Science and Technology, 7028 Trondheim, Norway; aleksandr.ianevski@ntnu.no (A.I.); rouany@stud.ntnu.no (R.Y.); svetlana.biza@yandex.ru (S.B.); Hilde.Lysvand@ntnu.no (H.L.); kirsti.loseth@ntnu.no (K.L.); veslemoy.m.landsem@ntnu.no (V.M.L.); valentyn.oksenych@ntnu.no (V.O.); sten.e.erlandsen@ntnu.no (S.E.E.); per.a.aas@ntnu.no (P.A.A.); lars.hagen@ntnu.no (L.H.); caroline.h.pettersen@ntnu.no (C.H.P.); jan.afset@ntnu.no (J.E.A.); svein.a.nordbo@ntnu.no (S.A.N.); magnar.bjoras@ntnu.no (M.B.); 2Department of Medical Microbiology, St. Olavs Hospital, 7006 Trondheim, Norway; Mona.Hoyseter.Fenstad@stolav.no (M.H.F.); Janne.Fossum.Malmring@stolav.no (J.F.M.); 3Department of Immunology and Transfusion Medicine, St. Olavs Hospital, 7006 Trondheim, Norway; 4Institute of Technology, University of Tartu, 50090 Tartu, Estonia; Eva.Zusinaite@ut.ee (E.Z.); tanel.tenson@ut.ee (T.T.); 5Institute of Genomics Core Facility, University of Tartu, 51010 Tartu, Estonia; tuuli.reisberg@ut.ee

**Keywords:** antivirals, broad-spectrum antivirals, antiviral drug combinations

## Abstract

As of June 2020, the number of people infected with severe acute respiratory coronavirus 2 (SARS-CoV-2) continues to skyrocket, with more than 6.7 million cases worldwide. Both the World Health Organization (WHO) and United Nations (UN) has highlighted the need for better control of SARS-CoV-2 infections. However, developing novel virus-specific vaccines, monoclonal antibodies and antiviral drugs against SARS-CoV-2 can be time-consuming and costly. Convalescent sera and safe-in-man broad-spectrum antivirals (BSAAs) are readily available treatment options. Here, we developed a neutralization assay using SARS-CoV-2 strain and Vero-E6 cells. We identified the most potent sera from recovered patients for the treatment of SARS-CoV-2-infected patients. We also screened 136 safe-in-man broad-spectrum antivirals against the SARS-CoV-2 infection in Vero-E6 cells and identified nelfinavir, salinomycin, amodiaquine, obatoclax, emetine and homoharringtonine. We found that a combination of orally available virus-directed nelfinavir and host-directed amodiaquine exhibited the highest synergy. Finally, we developed a website to disseminate the knowledge on available and emerging treatments of COVID-19.

## 1. Introduction

Every year, emerging and re-emerging viruses such as SARS-CoV-2, SARS-CoV, Middle East respiratory syndrome coronavirus (MERS-CoV), Zika virus (ZIKV), Ebola virus (EBOV), influenza A virus (FLUAV) and Rift Valley fever virus (RVFV) surface from natural reservoirs to infect people [1,2]. As of June 2020, the number of people infected with SARS-CoV-2 continues to rise, with more than 6.7 million cases worldwide.

Coronaviruses (CoV) are a broad family of viruses, which include species such as SARS-CoV-2, SARS-CoV, MERS-CoV, HCoV-229E, HCoV-OC43, HCoV-NL63 and HCoV-HKU1. CoV virions are composed of single-stranded positive-sense RNA, a lipid envelope and several proteins, including the spike (S), envelope (E), membrane (M) and nucleocapsid (N). HCoV-229E, HCoV-OC43, HCoV-NL63 and HCoV-HKU1 are usually associated with mild, self-limiting upper respiratory tract infections like the common cold. By contrast, SARS-CoV-2, MERS-CoV or SARS-CoV infections can lead to serious diseases and death. Considering the current global pandemic, both the WHO and UN has highlighted the need for a better control of SARS-CoV-2 infections; however, developing novel virus-specific vaccines, monoclonal antibodies and antiviral drugs against SARS-CoV-2 can be time-consuming and costly [3,4,5,6,7,8,9,10]. Convalescent plasma and safe-in-man broad-spectrum antivirals (BSAAs) are readily available treatment options that can circumvent these difficulties.

Human convalescent plasma is collected from recently recovered individuals and is used to transfer passive antibody immunity to those who have recently been infected or have yet to be exposed to the virus. However, the reliability of diagnostic assays, as well as the quantity and neutralizing capacity of antibodies in the plasma, should be considered. Both the WHO and European Centre for Disease Prevention and Control (ECDC) have recommended the evaluation of convalescent plasma for the prevention and treatment of the COVID-19 disease in controlled trials (WHO/HEO/R&D Blueprint (nCoV)/2020.1) [11]. Indeed, several trials have shown that convalescent plasma reduced the viral load and was safe for the treatment of patients with COVID-19 [12,13]. Interestingly, the studies also revealed a variability in the specific antibody production that was related to the degree of the symptomatic disease. Typically, IgM and IgG antibodies developed between 6–15 days after disease onset, and nearly all patients were seropositive within three weeks. Some recovered patients did not have high titers of neutralizing antibodies, possibly suggesting that antibody levels declined with time and/or that only a subset of patients produced high-titer responses. It is also possible that non-neutralizing antibodies and cellular responses not connected to antibody production were contributing to the protection and recovery, as described for other viral diseases [14,15,16].

BSAAs are small molecules that inhibit human viruses belonging to two or more viral families and that have passed the first phase of clinical trials. We have recently reviewed and summarized the information on dozens of safe-in-man BSAAs in the freely accessible database (https://drugvirus.info/) [17,18,19]. Forty-six of these agents inhibited SARS-CoV, MERS-CoV, HCoV-229E, HCoV-OC43, HCoV-NL63 and/or HCoV-HKU1. Additionally, clinical investigations have started recently into the effectiveness of BSAAs such as lopinavir, ritonavir, remdesivir, hydroxychloroquine and arbidol against COVID-19 (NCT04252664, NCT0425487, NCT04255017, NCT04261517 and NCT04260594). Such BSAAs represent a promising source of drug candidates for SARS-CoV-2 prophylaxis and treatment.

Here, we report the isolation of seven different SARS-CoV-2 strains from COVID-19 patients in Norway. We showed that UVC and high temperatures destroyed SARS-CoV-2, establishing a rationale and methodology for safe work in the laboratory. We developed a neutralization assay using SARS-CoV-2 virus and monkey Vero-E6 cells and identified the most potent sera from recovered patients for the treatment of SARS-CoV-2-infected patients. We also screened 136 safe-in-man broad-spectrum and identified the anti-SARS-CoV-2 activity of salinomycin, nelfinavir, amodiaquine, obatoclax, emetine and homoharringtonine in vitro. We further showed that combinations of virus-directed drug nelfinavir and host-directed drugs, such as salinomycin, amodiaquine, obatoclax, emetine and homoharringtonine, act synergistically. Finally, we developed a website summarizing the scientific and clinical information regarding available and emerging anti-SARS-CoV-2 options.

## 2. Materials and Methods

### 2.1. Patient Samples

We selected subjects among healthy individuals, in- and outpatients, as well as patients recovered from SARS-CoV-2 or endemic coronavirus infections. Hospitalization was determined by whether a patient was able to manage symptoms effectively at home, according to local guidelines. ICU admittance was evaluated consistently with the WHO interim guidance on “Clinical management of severe acute respiratory infection when COVID-19 is suspected” (WHO/2019-nCoV/clinical/2020.4). The patients gave their informed consent through the koronastudien.no website. For ICU patients, consent for sample collection was received after treatment or from relatives. For children, consent for sample collection was received from their parents. Donors were recruited through information at the blood collection centers websites and through national media. Nasopharyngeal swabs (NPS) and blood samples were collected. All patients were treated in accordance with good clinical practice, following study protocols. The study was approved by the national ethical committee (clinical trial: NCT04320732; REK: 124170).

### 2.2. Cell Cultures

Human telomerase reverse transcriptase-immortalized retinal pigment epithelial (RPE), lung adenocarcinoma A427, non-small-cell lung cancer Calu-3 and epithelial colorectal adenocarcinoma Caco-2 cells were grown in DMEM-F12 supplemented with 100 μg/mL streptomycin and 100 U/mL penicillin (Pen-Strep), 2 mM l-glutamine, 10% FBS and 0.25% sodium bicarbonate (Sigma-Aldrich, St. Louis, MO, USA). Human neural progenitor cells derived from iPS cells were generated and maintained as described previously [20]. Human large-cell lung carcinoma NCI-H460, colon cancer SW620, colorectal carcinoma SW480 and HT29 cells were grown in RPMI medium supplied with 10% FBS and Pen-Strep. Human adenocarcinoma alveolar basal epithelial A549, human embryonic kidney HEK-293 cells and kidney epithelial cells extracted from an African green monkey (Vero-E6) were grown in DMEM supplemented with 10% FBS and Pen-Strep. The cell lines were maintained at 37 °C with 5% CO_2_.

### 2.3. Virus Isolation

The SARS-CoV-2 strains were isolated and propagated in a Biological Safety Level 3 (BSL-3) facility. Two hundred microliters of nasopharyngeal swabs (NPS) in universal transport medium were diluted 5 times in culture medium (DMEM) supplemented with 0.2% bovine serum albumin (BSA), 0.6 μg/mL penicillin, 60 μg/mL streptomycin and 2 mM l-glutamine and inoculated into Vero-E6 cells. After 4 days of incubation, the media were collected, and the viruses were passaged once again in Vero-E6 cells. After 3 days, a clear cytopathic effect (CPE) was detected, and the virus culture was harvested. Virus concentration was determined by RT-qPCR and plaque assays.

### 2.4. Virus Detection and Quantification

Viral RNA was extracted using the NTNU_MAG_V2 protocol, a modified version of the BOMB-protocol [21]. The eluate (2.5 or 5 μL) was analyzed by RT-qPCR using a CFX96 Real-Time Thermal Cycler (Bio-Rad, Hercules, CA, USA) as described elsewhere [22], with some modifications. One 20-μL reaction contained 10 μL of QScript XLT One-Step RT-qPCR ToughMix (2×) (Quanta BioSciences, Beverly, MA, USA), 1 μL each of the primer and probe with final respective concentrations of 0.6 and 0.25 μM and 2 μL of molecular-grade water. Thermal cycling was performed at 50 °C for 10 min for reverse transcription, followed by 95 °C for 1 min and then 40 cycles of 95 °C for 3 s and 60 °C for 30 s.

For testing the production of infectious virions, we titered the viruses as described in our previous studies [23,24,25]. In summary, media from the viral culture were serially diluted from 10^−2^ to 10^−7^ in serum-free DMEM containing 0.2% bovine serum albumin. The dilutions were applied to a monolayer of Vero-E6 cells in 6 or 24-well plates. After one hour, cells were overlaid with virus growth medium containing 1% carboxymethyl cellulose and incubated for 96 h. The cells were fixed and stained with crystal violet dye, and the plaques were calculated in each well and expressed as plaque-forming units per mL (pfu/mL).

### 2.5. Viral Genome Sequencing

Viral RNA was extracted using the NTNU_MAG_V2 protocol, a modified version of the BOMB-protocol. Libraries were prepared using a NuGEN Trio RNA-seq kit. Sequencing was performed on the NextSeq500 instrument (NS500528; setup: PE 2 × 75 bp + single index (8 bp)) using a NextSeq MID150 sequencing kit and NextSeq MID flow cell (NCS version: 2.2.0.4). Reads were aligned using the Bowtie 2 software package version 2.3.4.1 to the reference viral genome Wuhan-Hu-1/2019. Sequence alignments were converted to binary alignments and sorted using SAMtools version 1.5. The consensus FASTQ sequences were obtained with bcftools and vcfutils.pl (from SAMtools) and converted to FASTA using seqtk (https://github.com/lh3/seqtk). Viral genomes in FASTA formats were submitted to www.gisaid.org. The accession numbers of these genomes are: EPI_ISL_450352 (hCoV-19/Norway/Trondheim-E10/2020), EPI_ISL_450351 (hCoV-19/Norway/Trondheim-E9/2020), EPI_ISL_450350 (hCoV-19/Norway/Trondheim-S15/2020), EPI_ISL_450349 (hCoV-19/Norway/Trondheim-S12/2020), EPI_ISL_450348 (hCoV-19/Norway/Trondheim-S10/2020), EPI_ISL_450347 (hCoV-19/Norway/Trondheim-S5/2020) and EPI_ISL_450346 (hCoV-19/Norway/Trondheim-S4/2020). Transmission of the strains was visualized using https://nextstrain.org/ncov/global?f_author=Aleksandr%20Ianevski%20et%20al.

### 2.6. UVC and Thermostability Assays

The virus (multiplicity of infection, moi 0.1) was aliquoted in Eppendorf tubes and incubated at −80, −20, 4, 20, 37 and 50 °C for 48 h or at 96 °C for 10 min. Alternatively, the virus was aliquoted in wells of a 96-well plate without a lid. The virus was exposed to UVC light (λ = 254 nm, ≥125 μW/cm^2^) for 10, 20, 40, 80, 160, 320 and 640 s using a germicidal lamp in a biosafety cabinet. The thermo and UVC stability of the viral RNA was analyzed by RT-qPCR. Subsequently, Vero-E6 cells were infected with the virus for 72 h, and cell viability was measured using a CellTiter-Glo assay (Promega, Madison, WI, USA). Luminescence was read using a plate reader.

### 2.7. Neutralization Assay

Approximately 4 × 10^4^ Vero-E6 cells were seeded per well in 96-well plates. The cells were grown for 24 h in DMEM supplemented with 10% FBS and Pen-Strep. Serum samples were diluted 100-fold, and protein concentrations were quantified using NanoDrop. Serum samples were prepared in 3-fold dilutions at 7 different concentrations, starting from 1 mg/mL in the virus growth medium (VGM) containing 0.2% BSA and Pen-Strep in DMEM. Virus hCoV-19/Norway/Trondheim-E9/2020 was added to the samples to achieve a moi of 0.1 and incubated for 1h at 37 °C. 0.1% DMSO was added to the control wells. The Vero-E6 cells were incubated for 72 h with VGM. After the incubation period, the medium was removed, and a CellTiter-Glo assay was performed to measure viability.

### 2.8. Drug Test

We have previously published a library of safe-in-man BSAAs [26]. Appendix A
Appendix A lists these and other potential BSAAs, their suppliers and catalogue numbers. To obtain 10-mM stock solutions, compounds were dissolved in dimethyl sulfoxide (DMSO; Sigma-Aldrich, Darmstadt, Germany) or milli-Q water. The solutions were stored at −80 °C until use.

Approximately 4 × 10^4^ Vero-E6 cells were seeded per well in 96-well plates. The cells were grown for 24 h in DMEM supplemented with 10% FBS and Pen-Strep. The medium was replaced with VGM containing 0.2% BSA and Pen-Strep in DMEM. The compounds were added to the cells in 3-fold dilutions at 7 or 8 different concentrations, starting from 30 μM. No compounds were added to the control wells. The cells were mock- or hCoV-19/Norway/Trondheim-E9/2020-infected at a moi of 0.1. After 72 h of infection, the medium was removed from the cells, and a CellTiter-Glo assay was performed to measure viability.

### 2.9. Drug and Serum Sensitivity Quantification

The half-maximal cytotoxic concentration (CC_50_) for each compound was calculated based on viability/death curves obtained on mock-infected cells after nonlinear regression analysis with a variable slope using GraphPad Prism software version 7.0a (GraphPad Software, San Diego, CA, USA). The half-maximal effective concentrations (EC_50_) were calculated based on the analysis of the viability of infected cells by fitting drug dose–response curves using the four-parameter (4PL) logistic function *f*(*x*):(1)$$f\left(x\right)=A_{min}+\frac{A_{max}-A_{min}}{1+{\left(\frac{x}{m}\right)}^{\lambda }},$$
where *f*(*x*) is a response value at dose *x*, *A_min_* and *A_max_* are the upper and lower asymptotes (minimal and maximal drug effects), *m* is the dose that produces the half-maximal effect (EC_50_ or CC_50_) and *λ* is the steepness (slope) of the curve. A relative effectiveness of the drug was defined as the selectivity index (SI = CC_50_/EC_50_). The threshold of the SI used to differentiate between active and inactive compounds was set to 3.

Area under the dose-response curve AUC was quantified as:(2)$$\mathrm{AUC}=\int_{x_{min}}^{x_{max}}f\left(x\right)dx,$$
using the numerical integration implemented in the MESS R package (Bell Laboratories, Murray Hill, NJ, USA), where $x_{max}$ and $x_{min}$ are the maximal and minimal measured doses. Serum sensitivity score (SSS) was quantified as a normalized version of the standard AUC (with the baseline noise subtracted and normalization of the maximal response at the highest concentration often corresponding to off-target toxicity) as
(3)$$\mathrm{SSS}=\frac{\mathrm{AUC}-t\left(x_{max}-x_{min}\right)}{\left(100-t\right)\left(x_{max}-x_{min}\right)\log_{10}A_{min}},$$
where activity threshold *t* equals 10%.

### 2.10. Drug Combination Test and Synergy Calculations

Vero-E6 cells were treated with different concentrations of a combination of two BSAAs. After 72 h, cell viability was measured using a CellTiter-Glo assay. To test whether the drug combinations acted synergistically, the observed responses were compared with expected combination responses. The expected responses were calculated based on the ZIP reference model using SynergyFinder web application, version 2 [27].

### 2.11. ELISA Assays

We measured the IgG and IgM in human serum using Epitope Diagnostics enzyme-linked immunosorbent assays (ELISA) according to manufacturer specifications (Epitope Diagnostics, San Diego, CA, USA). Background-corrected optical density values were divided by the cutoff to generate signal-to-cutoff (s/co) ratios. Samples with s/co values greater than 1.0 were considered positive. The Pearson correlation coefficients were calculated by means of the stats R package, with the significance determined using a Student’s *t*-test.

### 2.12. Gene Expression Analysis

Vero-E6 cells were treated with nelfinavir, amodiaquine or both drugs at indicated concentrations. Cells were infected with the hCoV-19/Norway/Trondheim-E9/2020 strain at moi 0.1 or mock. After 24 h, total RNA was isolated using RNeasy Plus Mini kit (Qiagen, Hilden, Germany). Libraries were prepared and sequenced on a NextSeq500 (NS500528) instrument (set up: PE 2 × 75 bp + single index 8 bp) using a NextSeq MID150 sequencing kit, NextSeq MID flow cell, NCS version: 2.2.0.4. Reads were aligned using the Bowtie 2 software package version 2.3.4.1 to the NCBI reference sequence for SARS-CoV-2 (NC_045512.2) and to the human GRCh38 genome. Number of mapped and unmapped reads that aligned to each gene were obtained with the featureCounts function from Rsubread R-package version 2.10. The GFF table for the reference sequence was downloaded from https://ftp.ncbi.nlm.nih.gov/genomes/all/GCF/009/858/895/GCF_009858895.2_ASM985889v3/GCF_009858895.2_ASM985889v3_genomic.gff.gz and flattened to GTF format and given as an additional argument to the Rsubread function. The heatmaps were generated using the pheatmap package (https://cran.r-project.org/web/packages/pheatmap/index.html) based on log2-transformed profiling data.

### 2.13. Website

We reviewed the current landscape of the available diagnostic tools, as well as the emerging treatment and prophylactic options for the SARS-CoV-2 pandemic and have summarized the information in a database that can be freely accessed at https://sars-coronavirus-2.info. The information in the database was obtained from PubMed, clinicaltrials.gov, DrugBank, DrugCentral, the Chinese Clinical Trials Register (ChiCTR) and EU Clinical Trials Register databases [28,29,30], as well as other public sources. The website was developed with PHP v7 technology using D3.js v5 (https://d3js.org/) for visualization. The COVID-19 statistics are automatically exported from the COVID-19 Data Repository by the Center for Systems Science and Engineering (CSSE) at Johns Hopkins University (https://github.com/CSSEGISandData/COVID-19).

## 3. Results

### 3.1. Isolation of SARS-CoV-2 from COVID-19 Patients

We isolated seven SARS-CoV-2 strains from 22 NPS samples of COVID-19 patients using Vero-E6 cells. The RT-qPCR cycle threshold (Ct) values were 18–20 before and 13–15 after propagation of the viruses in Vero-E6 cells (Figure 1a,b). We sequenced seven strains and found that the sequences differed from the reference hCoV-19/Wuhan/WIV04/2019 strain by a few missense mutations (Figure 1c). Phylogenetic analysis showed a close relationship between the strains (Figure 1d). In cross-referencing our sequence data with the pathogen-tracking resource NextStrain.org, we determined that the SARS-CoV-2 strains isolated in Trondheim originated from China, Denmark, the USA and Canada (Figure 1e).

In addition, we tested several cell lines and found that Vero-E6 was the most susceptible for virus-mediated death and virus amplification (Appendix A). To establish the rationale for safe work, we incubated the virus at different temperatures for 48 h or exposed to UVC radiation for different time periods. The resulting virus preparations were analyzed by RT-qPCR and used to infect Vero-E6 cells. Virus incubation at 37 °C for 48 h or UVC exposure for 10 sec destabilized the virus and rescued infected cells from virus-mediated death (Appendix A).

### 3.2. Sera from Patients Recovered from COVID-19 Contain Antibodies That Neutralize SARS-CoV-2

We evaluated the neutralization capacity of seven serum samples obtained from patients recovered from COVID-19. We also used sera from patients recovered from endemic coronavirus infections and from healthy blood donors as controls. Five samples from patients recovered from COVID-19 had serum sensitivity scores (SSS) > 5 and rescued >50% cells from virus-mediated death at 1 mg/mL (Figure 2a,b). Notably, that serum from a recovered patient with SSS = 7.2 neutralized all seven SARS-CoV-2 strains (Appendix A).

Our neutralization test of 32 samples (Appendix A) showed a moderate positive correlation with IgG (r = 0.59, *p* < 0.001) and IgM (r = 0.43, *p* = 0.01) s/co values obtained using commercial ELISA kits that recognize the SARS-CoV-2 N protein (Figure 2c,d). However, the correlation between the IgG and IgM ELISA results was only r = 0.28, *p* = 0.11. Furthermore, we found a moderate negative correlation between SSSs and time intervals between the SARS-CoV-2 diagnosis and serum collection for 66 samples (−0.5, *p* < 0.025; Figure 2d). Altogether, these results suggest that patients diagnosed with COVID-19 produce different immune responses to the SARS-CoV-2 infection and that the neutralization capacity of convalescent sera declines with time.

Through the literature review, we made a database to summarize safe-in-man BSAAs (https://drugvirus.info/). Recently, we have expanded on the spectrum of activities for some of these agents [17,18,19,26,31]. Some of these agents could be repositioned for the treatment of a SARS-CoV-2 infection.

We tested 136 agents against SARS-CoV-2 in VERO-E6 cells. Remdesivir was included as a positive control [32] and nicotine as a negative control. Seven different concentrations of the compounds were added to virus-infected cells. Cell viability was measured after 72 h to determine compound efficiency. After the initial screening, we identified apilimod, emetine, amodiaquine, obatoclax, homoharringtonine, salinomycin, arbidol, posaconazole and nelfinavir as compounds that rescued virus-infected cells from death (AUC from 285 to 585; Appendix A). The compounds we identified possessed a structure-activity relationship (Figure 3a). AUC for remdesivir was 290. Interestingly, 10 μM of nicotine rescued cells from virus-mediated death but altered the cell morphology (AUC = 239; Appendix A).

We repeated the experiment with hit compounds, monitoring their toxicity and efficacy. We confirmed the antiviral activity of emetine, amodiaquine, obatoclax, homoharringtonine, salinomycin and nelfinavir (Figure 3b,c). Importantly, amodiaquine had a superior SI over its analogs chloroquine, hydroxychloroquine, quinacrine and mefloquine (Appendix A). Thus, we identified and validated anti-SARS-CoV-2 activities for six BSAAs in Vero-E6 cells.

### 3.3. BSAA Combinations Are Effective against the SARS-CoV-2 Infection

To test for potential synergism among the validated hits, we treated cells with varying concentrations of a two-drug combination and monitored the cell viability (Figure 4a). The observed drug combination responses were compared with the expected combination responses calculated by means of the zero-interaction potency (ZIP) model [27,33]. We quantified synergy scores, which represent the average excess response due to drug interactions (i.e., 10% of cell survival beyond the expected additivity between single drugs has a synergy score of 10). We found that combinations of nelfinavir with salinomycin, amodiaquine, homoharringtonine and obatoclax, as well as the combination of amodiaquine and salinomycin, were synergistic (most synergistic area scores >10; Figure 4b). Moreover, the nelfinavir-amodiaquine treatment was effective against all seven SARS-CoV-2 strains (Figure 4c). Thus, we identified synergistic drug combinations against SARS-CoV-2 infections.

We next profiled transcriptional responses to nelfinavir, amodiaquine or both drugs in virus- or mock-infected Vero-E6 cells at 24 h. We showed that the addition of nontoxic but effective concentrations of drugs slightly affected the transcription of immune-related genes in virus-infected cells (Appendix A). These genes (*CXCL1, CXCL2, CXCL3, CXCL8, CXCL10, CXCL11, OASL, IFNL1, MX1* and *HERC5)* are needed for alarming neighboring cells about the ongoing infection and for the protection of the organism from repeated infections. Amodiaquine and its combination with nelfinavir lowered the transcription of viral genomic and sub-genomic RNAs (Appendix A).

### 3.4. Sars-Coronavirus-2.info Website Summarizes Emerging Antiviral Options

To rapidly respond to the COVID-19 outbreak, we developed a freely accessible website summarizing novel anti-SARS-CoV-2 developments and currently approved diagnostic options around the globe. It also tracks the development of therapeutic/antiviral drugs and vaccines.

The “Treatment” section of the website summarizes 542 in-progress and completed clinical trials that test the efficacy of therapeutic agents to treat COVID-19 or complications that arise from COVID-19. These trials include over 192 unique therapeutic agents in varying combinations and applications. Importantly, we list 71 clinical trials that are already completed or are projected to be completed by the end of June 2020. Of note, among these are trials of remdesivir, favipiravir, lopinavir/ritonavir, hydroxychloroquine, dipyridamole and interferons alpha and beta, which are all phase 3 or 4 clinical trials scheduled to be currently completed.

The “Prevention” section summarizes 23 current vaccine trials taking place around the globe. Although vaccine development lags considerably behind drug development, several repurposed vaccine options have also emerged. This includes trials of the cross-reactivity of the MMR (Measles, Mumps and Rubella) vaccine, as well as several trials of the Bacillus Calmette–Guérin (BCG) vaccine among high-risk populations, such as healthcare workers.

Finally, the “Testing” section of the website provides a summary of 377 currently available laboratory-based and point-of-care diagnostic tests that are approved for clinical diagnosis in at least one country.

The website also includes predictions of experimental and approved drugs effective against SARS-CoV-2, as well as provides a summary of information about the coronavirus pandemic. The website allows interactive exploration of the data with built-in feedback and is available in several languages. The website is updated as soon as novel anti-SARS-CoV-2 options emerge, or the statuses of existing ones are updated.

## 4. Discussion

Here, we reported the isolation of seven SARS-CoV-2 strains from samples of patients suffering from COVID-19. Full-genome sequencing revealed that the strains were highly similar (98.8%) to one another and to the strains circulating in China, Denmark, the USA and Canada. All seven strains contain D614G in the S protein. Strains with this mutation began spreading in Europe in early February and became dominant in other regions (https://doi.org/10.1101/2020.04.29.069054).

We screened 12 cell lines for their susceptibility for the SARS-CoV-2 infection and virus replication. Vero-E6 cells appeared to be the most susceptible cell line for virus-mediated cell death and virus propagation. This cell line has been widely used in toxicology, virology and pharmacology research, as well as in the production of vaccines and diagnostic reagents. The cell line is interferon-deficient; i.e., unlike normal mammalian cells, it does not secrete interferon alpha or beta when infected by viruses [34]. Moreover, this cell line was used routinely in anti-SARS-CoV-2 research [35,36,37,38].

We also showed that >10 sec of UVC radiation or 48 h incubation at 37 °C neutralized SARS-CoV-2, establishing a rationale and methodology for safe work in the laboratory. These results are consistent with previous studies showing that physical factors destabilize SARS-CoV-2 and other viruses [39,40,41,42].

Neutralization tests are crucial tools for the assessment of previous SARS-CoV-2 exposure and potential immunity [12,13]. We have developed a test to assess the neutralization capacity of serum samples from patients recovered from SARS-CoV-2 infections, patients with endemic coronavirus infections and healthy blood donors. Our results suggest that COVID-19 patients respond differently to the SARS-CoV-2 infection. Moreover, the neutralization capacity of convalescence sera declined with time. Thus, the neutralization test allowed us to identify the most potent sera from patients recovered from COVID-19 for the treatment of SARS-CoV-2-infected patients.

Moreover, results from our neutralization test positively correlated with those from commercial ELISA assays. However, the correlation between the IgG and IgM ELISA results was only moderate. The difference could be associated with the time of the sample collection, production of the immunoglobulins or sensitivity that can be attributed to the technique and the antigen in use (i.e., IgM is the first immunoglobulin to be produced in response to an antigen and can be detected during early onset of disease, whereas IgG is maintained in the body after initial exposure for the long-term response and can be detected after the infection has passed).

There were certain concerns regarding the antibody-dependent cell-mediated toxicity of convalescent sera [43]. However, the recent safety study on 5000 hospitalized patients transfused under the U.S. Food and Drug Administration’s national Expand Access Program for COVID-19 revealed no toxicity (https://doi.org/10.1101/2020.05.12.20099879).

Drug repurposing, also called drug repositioning, is a strategy for generating an additional value from an existing drug by targeting diseases other than that for which it was originally intended [44,45]. This has significant advantages over new drug discoveries, since chemical synthesis steps, manufacturing processes, reliable safety and pharmacokinetic properties have already been studied in preclinical (animal model) and early clinical developmental phases (phase 0, I and IIa). Therefore, drug repositioning for COVID-19 provides unique translational opportunities, including a substantially higher probability of success to the market as compared with developing new virus-specific drugs and vaccines, as well as significantly reduced costs and timelines to clinical availability [26,46,47].

We tested 136 safe-in-man BSAAs against SARS-CoV-2 in cell cultures. We identified salinomycin, obatoclax, amodiaquine, nelfinavir, emetine and homoharringtonine as having anti-SARS-CoV-2 activity, which we put forward as potential anti-SARS-CoV-2 drug candidates.

Nelfinavir (Viracept) is an orally bioavailable inhibitor of human immunodeficiency virus HIV-1 (750 mg per os (PO) q8hr). It targets HIV protease for the treatment of HIV infections [48]. Molecular docking studies predict that nelfinavir binds to the SARS-CoV-2 protease [49]. Nelfinavir could also inhibit cell fusion caused by the SARS-CoV-2 S glycoprotein (https://doi.org/10.1101/2020.04.06.026476) [50]. It also inhibits Chikungunya virus (CHIKV), Dengue virus (DENV), hepatitis C virus (HCV), herpes simplex virus 1 (HSV-1) and SARS-CoV infections (https://doi.org/10.1039/C5RA14469H) [51,52,53].

Amodiaquine is a medication used to treat malaria. The recommended dose for a course of amodiaquine is 30 mg amodiaquine base/kg body weight over three days, i.e., 10 mg/kg/day [54]. It also shows broad-spectrum activity against ZIKV, DENV, HCV, MERS-CoV, SARS-CoV, SARS-CoV-2, Ross River virus (RRV), Sindbis virus (SINV), West Nile virus (WNV), yellow fever virus (YFV), EBOV, Lassa virus (LASV), rabies virus (RABV), vesicular stomatitis virus (VSV) and HSV-1 viruses (https://doi.org/10.1101/2020.03.25.008482) [55,56,57,58,59,60]. Importantly, amodiaquine showed more potent antiviral activity than its analogs chloroquine and hydroxychloroquine.

Obatoclax was originally developed as an anticancer agent. Several phase II clinical trials have investigated the use of obatoclax in the treatment of leukemia, lymphoma, myelofibrosis and mastocytosis. A continuous 24 h infusion of obatoclax 25–60 mg/day for three days in two-week cycles or 3 h infusions in a 3-day cycle have previously been evaluated in cancer patients [61]. In addition, obatoclax showed antiviral activity against FLUAV, ZIKV, WNV, YFV, SINV, Junín Virus (JUNV), LASV, herpes simplex virus 2 (HSV-2), echovirus 1 (EV1), human metapneumovirus (HMPV), RVFV and lymphocytic choriomeningitis virus (LCMV) in vitro [18,24,25,62,63]. It was shown that obatoclax inhibited the viral endocytic uptake by targeting the cellular Mcl-1 protein [24].

Emetine is an antiprotozoal drug. It is administered by intramuscular or deep subcutaneous injection in a dose of 1 mg/kg/day (maximum 60 mg/day) for 10 days [64]. Emetine is also used to induce vomiting. In addition, it possesses antiviral effects against ZIKV, EBOV, RABV, cytomegalovirus (CMV), HCoV-OC43, HSV-2, EV1, HMPV, RVFV, FLUAV, HIV-1 and SARS-CoV-2 [17,18,36,65,66,67,68,69] (https://doi.org/10.1101/2020.03.25.008482). Emetine was proposed to inhibit viral polymerases, though it could have some other targets [70].

Homoharringtonine is an anticancer drug that is indicated for the treatment of chronic myeloid leukemia (2 mg/m^2^ IV daily × 7). It also possesses antiviral activities against hepatitis B virus (HBV), MERS-CoV, HSV-1, EV1, VZV and SARS-CoV-2 in vitro [18,36,71,72,73,74]. Homoharringtonine binds to the 80S ribosome and inhibits viral protein synthesis by interfering with the chain elongation [72].

Salinomycin is an orally bioavailable antibiotic that is used against Gram-positive bacteria in animal husbandry (0.2 mg/kg body weight (BW) PO). It also inhibits FLUAV, respiratory syncytial virus (RSV) and CMV infections [75,76]. Salinomycin was proposed to disrupt the endosomal acidification and to block entry of the viruses into cells [77,78].

Our results have uncovered several existing BSAAs that could be repositioned to SARS-CoV-2 infections. Since pharmacokinetic/pharmacodynamic and toxicology studies have already been performed on these compounds, in vivo efficacy studies could be initiated immediately, saving time and resources.

Combination therapies became a standard for the treatment of HIV and HCV infections. The reasons for using combinations rather than single antiviral are better efficacy, decreased toxicity and the prevention of resistance emergence. Here, we found that combinations of nelfinavir with salinomycin, amodiaquine, obatoclax, emetine or homoharringtonine were synergistic against SARS-CoV-2 in Vero-E6 cells. Interestingly, synergistic interactions occurred between compounds belonging to different SAR clusters (i.e., nelfinavir belongs to a separate cluster than amodiaquine or obatoclax-emetine-homoharringtonine). In particular, the synergy was achieved when a virus-directed drug was combined with host-directed ones. This observation agrees with other studies on such combinations and virus-host interactions [18,25,79,80] (https://doi.org/10.1101/2020.04.14.039925).

According to the available pharmacological data for these drugs, the most potent combination could be a combination of orally available nelfinavir and amodiaquine. This was also the combination that exhibited the highest synergy of all the drug combinations we tested, with the synergy score at the most synergistic area being 16.48 (i.e., 16.48% of cell survival beyond the expected additivity between the single drugs). There are no guidelines of what is considered a good synergy, but it is very common to consider synergy >10 as true (significant) synergy. Thus, the amodiaquine and nelfinavir combination could result in better efficacy and decreased toxicity for the treatment of SARS-CoV-2 and perhaps other viral infections.

Our future goal is to complete preclinical studies and translate our findings into trials in patients. The most effective and tolerable BSAAs or their combinations will have a global impact, improving the protection of the general population from emerging and re-emerging viral infections or coinfections and allowing the rapid management of drug-resistant strains. Our bigger ambition is to assemble a toolbox of BSAAs for the treatment of emerging and re-emerging viral infections. This toolbox can be offered to the WHO as a means for the fast identification of safe and effective antiviral options.

We have summarized the information about the status of currently available and emerging anti-SARS-CoV-2 options on the freely accessible website (https://sars-coronavirus-2.info). The website is updated regularly and incorporates novel anti-SARS-CoV-2 options as they emerge or changes the statuses of existing ones as updates occur.

## 5. Conclusions

In conclusion, sera from recovered patients, BSAAs and combinations of BSAAS, as well as other available and emerging treatments, could have pivotal roles in the battle against COVID-19 and other emerging and re-emerging viral diseases. Further development of these options could save time and resources that are required for the development of alternative virus-specific drugs and vaccines. This could have a global impact by decreasing morbidity and mortality, maximizing the number of healthy life years, improving the quality of life of infected patients and decreasing the costs of patient care curtailing to the impact of the current SARS-CoV-2 pandemic, as well as future viral outbreaks.

## Figures and Tables

**Figure 1 viruses-12-00642-f001:**
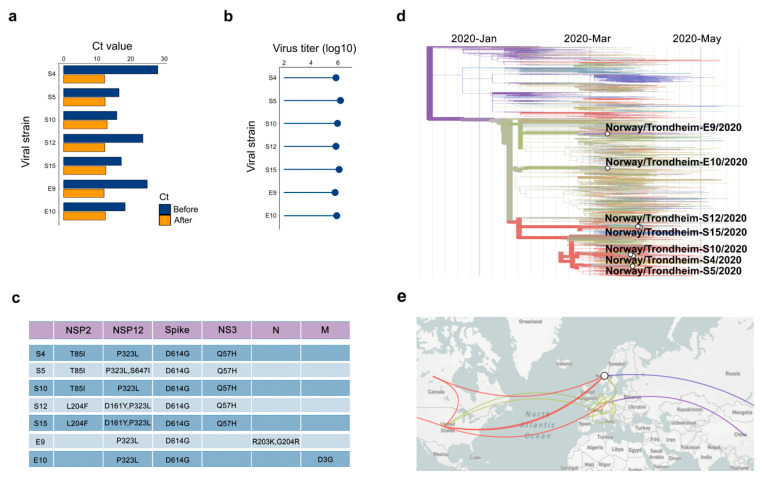
Isolation and characterization of 7 SARS-CoV-2 strains used in this study. (**a**) RT-qPCR analysis of 7 SARS-CoV-2 strains isolated from nasopharyngeal swabs. (**b**) Viruses amplified in Vero-E6 cells were quantified by plaque assay. (**c**) Table showing variations in amino acids between our SARS-CoV-2 strains and the reference hCoV-19/Wuhan/WIV04/2019 strain. (**d**) Phylogenetic analysis of 7 SARS-CoV-2 strains from Trondheim and other viral strains, which full-genome sequences were submitted to the GISAID database. (**e**) The origin of our 7 strains according to nextstrain.org.

**Figure 2 viruses-12-00642-f002:**
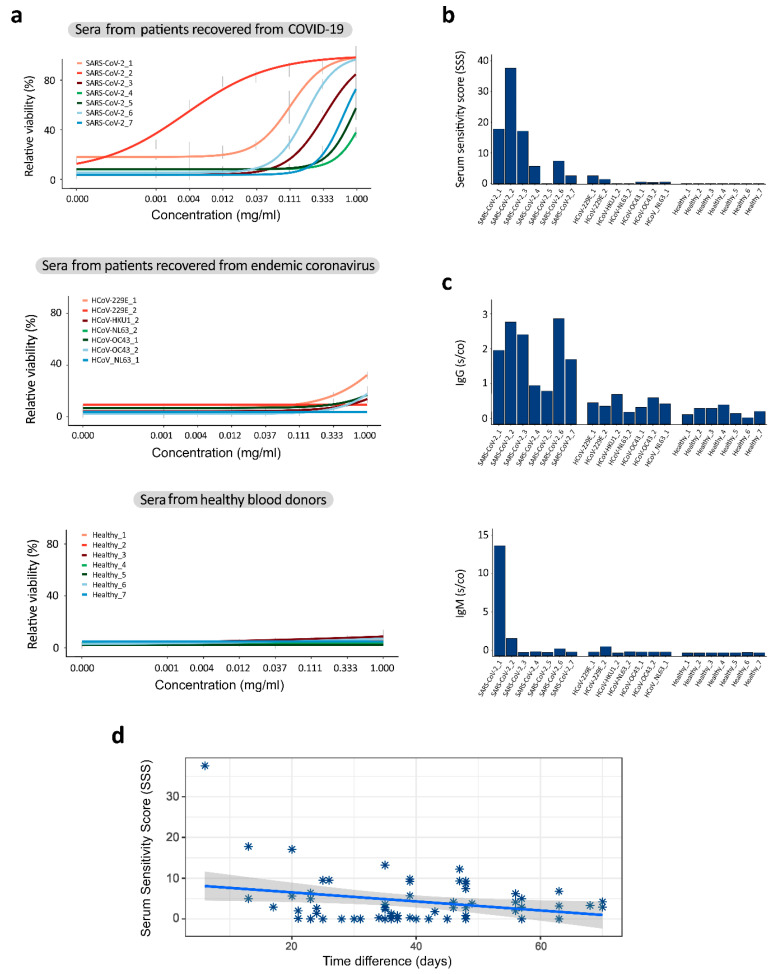
Sera from patients recovered from COVID-19 neutralized the SARS-CoV-2 virus and prevented the virus-mediated death of Vero-E6 cells. (**a**) The HCoV-19/Norway/Trondheim-E9/2020 strain (moi 0.1) was incubated with indicated concentrations of sera obtained from 7 patients recovered from COVID-19, 7 patients recovered from endemic coronavirus infections and 7 healthy blood donors. The mixtures were added to Vero-E6 cells. Cell viability was measured after 72 h. Mean ± SD, n = 3. (**b**) Serum sensitivity scores (SSS) were calculated based on curves in (**a**). (**b**,**c**) The IgG and IgM levels were analyzed in the sera of these patients using commercial Elisa kits. (**d**) Correlation analysis of serum sensitivity scores and time intervals between the SARS-CoV-2 diagnosis and sera collection.3.3. Repurposing Safe-In-Man BSAAs.

**Figure 3 viruses-12-00642-f003:**
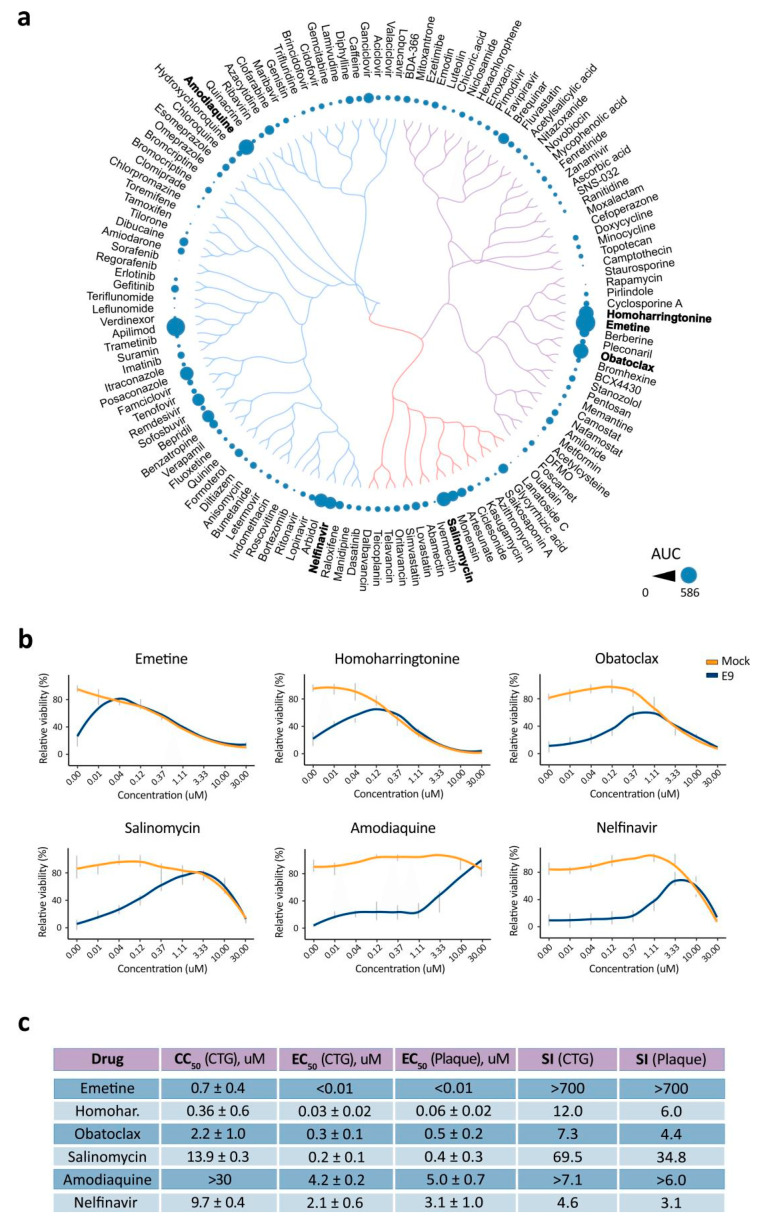
Anti-SARS-CoV-2 activity of safe-in man broad-spectrum antivirals in Vero-E6 cells. (**a**) Structure-antiviral activity relation of 136 broad-spectrum antivirals (BSAAs). The compounds were clustered based on their structural similarity calculated by ECPF4 fingerprints and visualized using the D3 JavaScript library. The anti-SARS-CoV-2 activity of the compounds was quantified using the AUC and shown as bubbles. Bubble size corresponds to compounds AUCs. (**b**) Vero-E6 cells were treated with increasing concentrations of a compound and infected with the HCoV-19/Norway/Trondheim-E9/2020 strain (moi, 0.1) or mock. After 72 h, the viability of the cells was determined using the CellTiter-Glo assay. Mean ± SD; n = 3. (**c**) Table showing half-maximal cytotoxic concentration (CC_50_), the half-maximal effective concentration (EC_50_) and selectivity indexes (SI = CC_50_/EC_50_) for selected anti-SARS-CoV-2 compounds calculated from CTG and plaque assays. Mean ± SD; n = 3.

**Figure 4 viruses-12-00642-f004:**
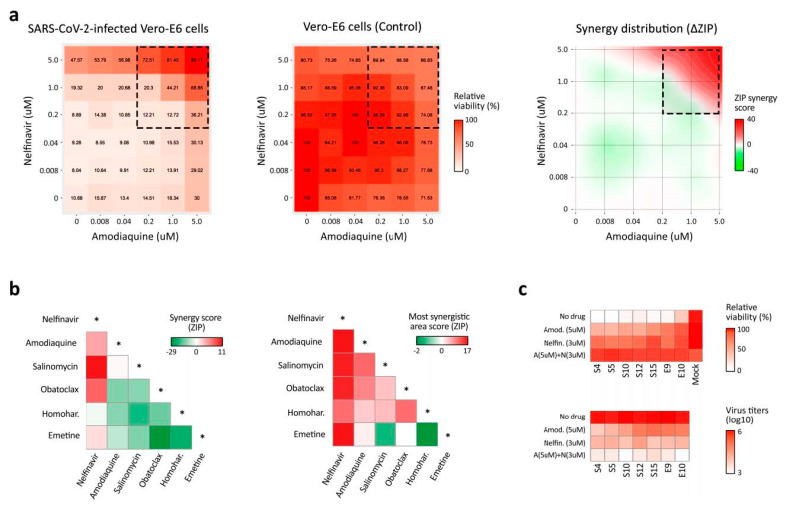
Effects of drug combinations on the SARS-CoV-2 infection. (**a**) The representative interaction landscapes of one of the drug combinations (amodiaquine-nelfinavir) measured using a CTG assay and SARS-CoV-2- and mock-infected cells (left and central panels). Synergy distribution is shown for virus-infected cells (right panel). (**b**) Synergy scores and the most synergistic area scores of 15 drug combinations. (**c**) The effects of the amodiaquine-nelfinavir combination on viral replication (lower panel), and the viability of cells infected with 7 different SARS-CoV-2 strains (upper panel).

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
