# Peer review of "Potential Antiviral Options against SARS-CoV-2 Infection"

_viruses, 2020, doi:10.3390/v12060642_

Round 1

Reviewer 1 Report

The work by Ianevski et al provides serum from recovered COVID-19 patients and broad-spectrum antivirals as options against SARS-CoV-2 infection. Combinational study between drugs were then performed in an effort to maximize the antiviral potency. The work in generally interesting to a broad audience. I would like to see a revised version of this manuscript after addressing the below concerns:

  1. Line 141: accession number of the viral genomes shall be provided.
  2. Line 180: the rationale to set SI=3 to differentiate the active and inactive compounds shall be provided.
  3. Line 190: the authors used cell viability as for drug synergy calculations without determine the live virus formation. This shall be repeated at least for the best combinations.
  4. Line 223, the author claimed sera from patients recovered from COVID-19 may provide beneficial advantage, which was supported by neutralizing assay in vitro. However, certain concerns in terms of the antibody-dependent cell-mediated cytotoxicity effect must be discussed when applied in vivo.
  5. Figure 2A, chloroquine and hydroxyl-chloroquine seems don’t work well in the authors’ experimental setting. This is strange because both drugs have been repeatedly reported to potently inhibit SARS-CoV-2 replication. This is the major concern that will compromise this whole study.
  6. Line 337, Discussion: provide essential pharmalogical data for the salinomycin, obatoclax, amodiaquine, nelfinavir, emetine, and homoharringtonine. For example their Cmax, half life, in vivo toxicity dose etc.
  7. As the situation of covid-19 changes rapidly, always update the case number and mortality if there is a chance of re-submission.

Author Response

Line 141: accession number of the viral genomes shall be provided.

Re: Many thanks for constructive feedback. Accession numbers are now provided: ”The accession numbers are EPI_ISL_450352 (hCoV-19/Norway/Trondheim-E10/2020), EPI_ISL_450351 (hCoV-19/Norway/Trondheim-E9/2020), EPI_ISL_450350 (hCoV-19/Norway/Trondheim-S15/2020), EPI_ISL_450349 (hCoV-19/Norway/Trondheim-S12/2020), EPI_ISL_450348 (hCoV-19/Norway/Trondheim-S10/2020), EPI_ISL_450347 (hCoV-19/Norway/Trondheim-S5/2020), EPI_ISL_450346 (hCoV-19/Norway/Trondheim-S4/2020)”. We also provided the detailed analysis of the origin of our strains.

Line 180: the rationale to set SI=3 to differentiate the active and inactive compounds shall be provided.

Re: We used 3-fold compound dilutions to calculate CC50, EC50, and SI in our studies (PMID: 31636628, 31635418, 29698664, etc.). We consider compounds with SI>3 as active.

Line 190: the authors used cell viability as for drug synergy calculations without determine the live virus formation. This shall be repeated at least for the best combinations.

Re: Fig. 4c shows the results of virus plaque assay of 7 different viruses produced in Vero-E6 cells treated with amodiaquine-nelfinavir combination (lower panel).

Line 223, the author claimed sera from patients recovered from COVID-19 may provide beneficial advantage, which was supported by neutralizing assay in vitro. However, certain concerns in terms of the antibody-dependent cell-mediated cytotoxicity effect must be discussed when applied in vivo.

Re: Indeed, there were certain concerns regarding the antibody-dependent cell-mediated toxicity of convalescent sera (PMID: 6605395). However, the recent safety study on 5,000 hospitalised patients transfused under the U.S. Food and Drug Administration`s national Expand Access Program for COVID-19 revealed no toxicity (https://doi.org/10.1101/2020.05.12.20099879).

Figure 2A, chloroquine and hydroxyl-chloroquine seems don’t work well in the authors’ experimental setting. This is strange because both drugs have been repeatedly reported to potently inhibit SARS-CoV-2 replication. This is the major concern that will compromise this whole study.

Re: We now added data on mefloquine to Fig. S4. By contrast to mefloquine and quinacrine, amodiaquine, chloroquine and hydroxychloroquine rescued cells from virus-mediated death and lowered virus titers produced in those cells. Amodiaquine had better selectivity, than chloroquine and hydroxychloroquine. A recent study showed that hydroxychloroquine and chloroquine have ‘no benefit’ for coronavirus patients and could even increase the risk of heart arrhythmias and mortality (https://doi.org/10.1016/S0140-6736(20)31180-6).

Line 337, Discussion: provide essential pharmacological data for the salinomycin, obatoclax, amodiaquine, nelfinavir, emetine, and homoharringtonine. For example their Cmax, half life, in vivo toxicity dose etc.

Re: We now provided some pharmacological data for obatoclax, amodiaquine, nelfinavir, emetine, and homoharringtonine in the discussion section of the manuscript:

“Nelfinavir (Viracept) is an orally bioavailable inhibitor of human immunodeficiency virus HIV-1 (750 mg PO q8hr).”

“The recommended dose for a course of amodiaquine is 30 mg amodiaquine base/kg body weight over 3 days, i.e., 10 mg/kg/day (PMID: 20065053).”

“Continuous 24-hour infusion of obatoclax 25-60 mg/day for 3 days in 2-week cycles or 3-hour infusion in 3 days cycle have previously been evaluated in cancer patient (PMID: 25285531).”

“Emetine is an antiprotozoal drug. It is administered by intramuscular or deep subcutaneous injection in a dose of 1 mg/kg/day (maximum, 60 mg/day) for 10 days (PMID: 4705155).”

“Homoharringtonine is an anticancer drug, which is indicated for treatment of chronic myeloid leukaemia (2 mg/m2 IV daily × 7).”

“Salinomycin is an orally-available antibiotic, which is used against Gram-positive bacteria in animal husbandry (maximum tolerable dose is 0.2 mg/kg BW, PO).”

As the situation of covid-19 changes rapidly, always update the case number and mortality if there is a chance of re-submission.

Re: We now updated the number of cases and mortality rate in the abstract and introduction.

Reviewer 2 Report

The manuscript written by Ianevski et al. focuses on screening of approved antivirals for their potential application against SARS-CoV-2 infection. The in vitro screening of their ability to supress cytotoxicity of the virus in Vero-E6 cells resulted in identification of several active compounds and some synergistic effects.

The topic of the article is very current and beneficial. It represents large set of data, is technically sound and in my opinion it is worth of publication in Viruses. However, there are some issues that deserve clarification.

Major Points:

The selection of cell lines for testing for coronavirus sensitivity is rather surprising. I understand that Vero cells were the most sensitive of the cell lines used. However, the authors should discuss why the kidney cells of an African green monkey i.e. Vero cells are more sensitive than lung epithelial cells which might be the target cells in vivo.

Line 145 the authors mention that: “The virus was exposed to UVC (λ = 254 nm)”; however, the information about the intensity W/cm2 is essential to provide valuable information about the virus sensitivity; also exposure time is indicated only in SFig 2 and not in the main body of the article.

Line 154: "The virus (Trondheim / S5 / 2020)" - it should be explained what is the origin of the virus. I assume it is one of the isolates. However, it should be explained why this particular strain was selected – the cytotoxicity?

Minor points:

Figure:  1 unify: Sera form patients recovered from   COVID-19 and SARS-CoV-2

Figure 3: I have been puzzled by the terms "Synergy scores" and "the most synergistic area scores" in my opinion, this deserves some clarification.

Line 115: the CPE abbreviation should be explained, add: cytopathic effect, please

Some typos:

Line 108: CO2 – 2 should be in a subscript

Line 118: "NTNU_MAG_V2" protocol I don't know what it is, maybe it would like a quote? Link to any log database?

Line 155: 37C should be changed to 37 °C

Lines 173 and 183: do not indicate the software manufacturer

In formula (3): minimum for absorbance in denominator should be a subscript

Lines 166 and 246: contradict the numbers of concentration points actually tested, in the methods it is stated 8, in the 7 and in Figure 2 they are 8 points in the graphs

Lines 342, 344, 352, 365, 384: they are present web links besides the reference numbers

Line 347: abbreviations of FLUAV, RSV and CMV viruses should be given in full (when first mentioned), this is important especially for RSV, where it can be either Respiratory syncytial virus of Rous sarcoma virus; this applies also for the viruses mentioned in lines 351-352, 360, 364, 369

Line 376: in vivo should be in italics

Is there any correlation of results in Figure 1 with those in Table S2 Results of neutralization and ELISA assays? Are some of the samples in S2 identical with those in Fig. 1 (differently named)?

Line 133: The authors report sequencing of viral genomes, but there is no information about the outcome of this in the paper (only that: Viral genomes were submitted to www.gisaid.org). I would expect some discussion about the sequencing data – interesting mutations etc. The authors might reconsider some exploitation of the data (however, this is optional…)

In conclusion, I recommend publication of the paper after addressing the most relevant points mentioned above.

Author Response

Major Points:

The selection of cell lines for testing for coronavirus sensitivity is rather surprising. I understand that Vero cells were the most sensitive of the cell lines used. However, the authors should discuss why the kidney cells of an African green monkey i.e. Vero cells are more sensitive than lung epithelial cells which might be the target cells in vivo.

Re: Vero-E6 cell line was the most susceptible for virus-mediated death and virus propagation. It has been widely used on toxicology, virology and pharmacology research, as well as on the production of vaccines and diagnostic reagents. They are interferon-deficient; unlike normal mammalian cells, they do not secrete interferon alpha or beta when infected by viruses (PMID 4302013). We now discuss this in the manuscript: “Vero-E6 cell line was the most susceptible for virus-mediated death and virus propagation. It has been widely used on toxicology, virology and pharmacology research, as well as, on the production of vaccines and diagnostic reagents. The cell line is interferon-deficient; unlike normal mammalian cells, it does not secrete interferon alpha or beta when infected by viruses (PMID 4302013).”

We also expanded the set of cell lines tested. We showed that lung epithelial Calu-3 and A427 cells were susceptible to virus-mediated death and virus propagation(Fig. S1).

Line 145 the authors mention that: “The virus was exposed to UVC (λ = 254 nm)”; however, the information about the intensity W/cm2 is essential to provide valuable information about the virus sensitivity; also exposure time is indicated only in Fig S2 and not in the main body of the article.

Re: We now indicated the intensity of UV lamp and exposure times in the text of the manuscript: “The virus was exposed to UVC (λ = 254 nm, ≥ 125 μW/cm2) for 10, 20, 40, 80, 160, 320 and 640 sec using germicidal lamp in a biosafety cabinet”.

Line 154: "The virus (Trondheim / S5 / 2020)" - it should be explained what is the origin of the virus. I assume it is one of the isolates. However, it should be explained why this particular strain was selected – the cytotoxicity?

Re: We sequenced the full-genomes of 7 isolates and provide accession numbers: ”The accession numbers are EPI_ISL_450352 (hCoV-19/Norway/Trondheim-E10/2020), EPI_ISL_450351 (hCoV-19/Norway/Trondheim-E9/2020), EPI_ISL_450350 (hCoV-19/Norway/Trondheim-S15/2020), EPI_ISL_450349 (hCoV-19/Norway/Trondheim-S12/2020), EPI_ISL_450348 (hCoV-19/Norway/Trondheim-S10/2020), EPI_ISL_450347 (hCoV-19/Norway/Trondheim-S5/2020),  EPI_ISL_450346 (hCoV-19/Norway/Trondheim-S4/2020)”.The genomes of the strains are highly similar (99.8%).  hCoV-19/Norway/Trondheim-E9/2020 is one of the earliest isolates, which we used throughout our study.

Minor points:

Figure:  1 unify: Sera form patients recovered from   COVID-19 and SARS-CoV-2

Re: Covid-19 is a disease, whereas SARS-CoV-2 is the virus. Therefore, we used both term in the figure legend: “Sera from patients recovered from Covid-19 neutralized SARS-CoV-2 virus and prevented virus-mediated death of Vero-E6 cells.”

Figure 3: I have been puzzled by the terms "Synergy scores" and "the most synergistic area scores" in my opinion, this deserves some clarification.

Re: The synergy score was calculated based on the 6 x 6 matrix of 2-drug combination. The most synergistic area score was calculated using 3 x 3 matrix. We now explain in more detail the synergy scores: “We found that combinations of nelfinavir with salinomycin, amodiaquine, homoharringtonine, and obatoclax, as well as combination of amodiaquine and salinomycin were synergistic (Fig. 4b, Most Synergistic Area scores > 10). This synergy score can be interpreted as the average excess response due to drug interactions (i.e., 10% of cell survival beyond expected additivity between single drugs).” and “In addition, amodiaquine and nelfinavir exhibited the highest synergy. The synergy score at most synergistic area was 16.48. This score can be interpreted as the average excess response due to drug interactions (i.e., 16.48% of cell survival beyond expected additivity between single drugs). There are no guidelines of what is considered a good synergy, but it is very common to consider synergy > 10 as true (significant) synergy. Thus, amodiaquine and nelfinavir combination could result in better efficacy and decreased toxicity for the treatment of SARS-CoV-2 and perhaps other viral infections.”

Line 115: the CPE abbreviation should be explained, add: cytopathic effect, please

Re: Corrected: “After 3 days a clear cytopathic effect (CPE) was detected, and virus culture was harvested.”

Some typos:

Line 108: CO2 – 2 should be in a subscript

Re: Corrected: “The cell lines were maintained at 37 °C with 5% CO2.”

Line 118: "NTNU_MAG_V2" protocol I don't know what it is, maybe it would like a quote? Link to any log database?

Re: This is a modified version of the BOMBprotocol (Oberacker et al., 2019), which is now undergoing patenting. For more information please visit: https://www.ntnu.edu/ntnu-covid-19-test.

Line 155: 37C should be changed to 37 °C

Re: Corrected: “The virus (hCoV-19/Norway/Trondheim-E9/2020) was added to the samples to achieve an moi of 0.1 and incubated for 1 h at 37 °C.”

Lines 173 and 183: do not indicate the software manufacturer

Re: We now indicated manufacturer for GraphPad Prism software version 7.0a (GraphPad Software, CA, USA) and MESS R package (Bell Laboratories, USA).

In formula (3): minimum for absorbance in denominator should be a subscript

Re: We now formatted minimum for absorbance in denominator in formula (3).

Lines 166 and 246: contradict the numbers of concentration points actually tested, in the methods it is stated 8, in the 7 and in Figure 2 they are 8 points in the graphs

Re: We performed initial screen using 7 drug concentrations. We used 8 different concentrations in the validation experiment because of emetine. We corrected the sentence accordingly: “The compounds were added to the cells in 3-fold dilutions at 7 or 8 different concentrations starting from 30 μM.”

Lines 342, 344, 352, 365, 384: they are present web links besides the reference numbers

Re: The links are for the preprints. As soon as the manuscripts are published, we will give a reference.

Line 347: abbreviations of FLUAV, RSV and CMV viruses should be given in full (when first mentioned), this is important especially for RSV, where it can be either Respiratory syncytial virus of Rous sarcoma virus; this applies also for the viruses mentioned in lines 351-352, 360, 364, 369

Re: We have now given abbreviations in full (when first mentioned).

Line 376: in vivo should be in italics

Re: Corrected: “This information could be used to initiate efficacy studies in vivo, saving time and resources.”

Is there any correlation of results in Figure 1 with those in Table S2 Results of neutralization and ELISA assays? Are some of the samples in S2 identical with those in Fig. 1 (differently named)?

Re: Yes, Table S2 contains information about Samples from Fig. 1. We renumbered some of them in Fig.1 for simplicity.

Line 133: The authors report sequencing of viral genomes, but there is no information about the outcome of this in the paper (only that: Viral genomes were submitted to www.gisaid.org). I would expect some discussion about the sequencing data – interesting mutations etc. The authors might reconsider some exploitation of the data (however, this is optional…)

Re: We sequenced the full genomes of 7 isolates. Table in Fig. 1c now shows variations in amino acids between our SARS-CoV-2 strains and the reference hCoV-19/Wuhan/WIV04/2019 strain. A panel in Fig. 1d also shows a phylogenetic analysis of 7 SARS-CoV-2 strains. We also state in the discussion that the strains have high sequence similarity.

In conclusion, I recommend publication of the paper after addressing the most relevant points mentioned above.

Re: Many thanks for your recommendation and comments.

Reviewer 3 Report

This reviewer doesn't see the novelty of this study.

The authors have screened a range of compounds to determine whether any have antiviral activity against SARS-CoV-2 in vitro. They identified several weak hits (only emetine has a reasonable SI to be taken forward - but then again they only explored toxicity in a single cell line which is a massive limitation of the study).

Rather than exploring the mechanism of lead hits or determining whether they work on particular viral proteins or if they have antiviral effect in vivo. The authors performed a synergy analysis which adds nothing significant to the manuscript. All the antagnostic interactions occur between compounds exist in the same SAR cluster indicated in figure 1, which is expected. Moreover, these synergy experiments were performed using compounds that have weak SI values - this seemed rather counterintuitive to include them in this type of analysis given they are essentially toxic in cell lines and thus are likely toxic in vivo.

Essentially there is no robust characterisation of the preliminary antiviral hits and this reviewer deems the study incomplete.

Author Response

The authors have screened a range of compounds to determine whether any have antiviral activity against SARS-CoV-2 in vitro.

Re: Apart from screening 136 compounds to determine their anti-SARS-CoV-2 activity:

  1. We isolated, sequenced and characterized 7 SARS-CoV-2 strains from Trondheim, Norway (Fig. 1). In cross-referencing our sequence data with the pathogen tracking resource, NextStrain.org, we determined that the SARS-CoV-2 strains isolated in Trondheim had originated from China, Denmark, USA and Canada (Fig. 1e).
  2. We established neutralization test of serum samples and showed that patients produced different responses to SARS-CoV-2 infection and that neutralization capacity of convalescence sera collected from patients diagnosed with COVID-19 declined with time (Fig. 2).
  3. Our results uncovered several existing BSAAs that could be re-positioned to SARS-CoV-2 infections. Because PK/PD and toxicology studies have been performed on these compounds, they could be used to initiate efficacy studies in vivo, saving time and resources. According to the available pharmacological data for these drugs, the most potent combination could be a combination of orally available drugs, nelfinavir and amodiaquine. This combination exhibited the highest synergy and could be taken to clinical trials.
  4. We have summarized the information about the status of currently available and emerging anti-SARS-CoV-2 options in a freely accessible website (https://sars-coronavirus-2.info).

They identified several weak hits (only emetine has a reasonable SI to be taken forward - but then again they only explored toxicity in a single cell line which is a massive limitation of the study).

Re: Four out of six compounds possessed their antiviral activity in sub-micromolar range in Vero-E6 cells. We screened 12 cell lines for susceptibility for SARS-CoV-2 infection and replication (Fig. S1). However, Vero-E6 appeared to be the most susceptible cell line for virus-mediated death and virus propagation (Fig. S1). This cell line has been widely used on toxicology, virology and pharmacology research, as well as, on the production of vaccines and diagnostic reagents. The cell line is interferon alpha or beta deficient (PMID 4302013). This cell line was used routinely in anti-SARS-CoV-2 research (PMID: 32251767, 32432977, 32149769, 32205232). We now discuss this issue in the manuscript.

Rather than exploring the mechanism of lead hits or determining whether they work on particular viral proteins or if they have antiviral effect in vivo, the authors performed a synergy analysis which adds nothing significant to the manuscript. All the antagnostic interactions occur between compounds exist in the same SAR cluster indicated in figure 1, which is expected.

Re: Synergistic interactions occurred between compounds belonging to different SAR clusters (i.e. nelfinavir belongs to separate cluster than amodiaquine or obatoclax-emetine-homoharringtonine).

Moreover, these synergy experiments were performed using compounds that have weak SI values - this seemed rather counterintuitive to include them in this type of analysis given they are essentially toxic in cell lines and thus are likely toxic in vivo.

Re: Both nelfinavir and amodiaquine from the lead combination are orally available FDA-approved drugs which make them attractive candidates for clinical trials against SARS-CoV-2 infections.

Essentially there is no robust characterisation of the preliminary antiviral hits and this reviewer deems the study incomplete.

Re: We now characterized antiviral hits and their combination in more detail. ´We performed transcriptomics analysis. We also provided pharmacological data for obatoclax, amodiaquine, nelfinavir, emetine, and homoharringtonine in the discussion section of the manuscript:

“Nelfinavir (Viracept) is an orally bioavailable inhibitor of human immunodeficiency virus HIV-1 (750 mg PO q8hr).”

“The recommended dose for a course of amodiaquine is 30 mg amodiaquine base/kg body weight over 3 days, i.e., 10 mg/kg/day (PMID: 20065053).”

“Continuous 24-hour infusion of obatoclax 25-60 mg/day for 3 days in 2-week cycles or 3-hour infusion in 3 days cycle have previously been evaluated in cancer patient (PMID: 25285531).”

“Emetine is an antiprotozoal drug. It is administered by intramuscular or deep subcutaneous injection in a dose of 1 mg/kg/day (maximum, 60 mg/day) for 10 days (PMID: 4705155).”

“Homoharringtonine is an anticancer drug, which is indicated for treatment of chronic myeloid leukaemia (2 mg/m2 IV daily × 7).”

“Salinomycin is an orally bioavailable antibiotic, which is used against Gram-positive bacteria in animal husbandry (maximum tolerable dose is 0.2 mg/kg BW, PO).”

Round 2

Reviewer 1 Report

Re: Indeed, there were certain concerns regarding the antibody-dependent cell-mediated toxicity of convalescent sera (PMID: 6605395). However, the recent safety study on 5,000 hospitalised patients transfused under the U.S. Food and Drug Administration`s national Expand Access Program for COVID-19 revealed no toxicity (https://doi.org/10.1101/2020.05.12.20099879).

But still, both reports shall be discussed.

Re: We now added data on mefloquine to Fig. S4. By contrast to mefloquine and quinacrine, amodiaquine, chloroquine and hydroxychloroquine rescued cells from virus-mediated death and lowered virus titers produced in those cells. Amodiaquine had better selectivity, than chloroquine and hydroxychloroquine. A recent study showed that hydroxychloroquine and chloroquine have ‘no benefit’ for coronavirus patients and could even increase the risk of heart arrhythmias and mortality (https://doi.org/10.1016/S0140-6736(20)31180-6).

The clinical study indicating ‘no benefit’ does not contradict with CQ and HCQ as a good positive control inhibitor in cell culture level.

Author Response

  1. We now discussed toxicity issue: "There were certain concerns regarding the antibody-dependent cell-mediated toxicity of convalescent sera [90]. However, the recent safety study on 5,000 hospitalised patients transfused under the U.S. Food and Drug Administration`s national Expand Access Program for COVID-19 revealed no toxicity (https://doi.org/10.1101/2020.05.12.20099879)."
  2. We agree that the clinical studies indicating ‘no benefit’ does not contradict with CQ and HCQ as a good positive control inhibitor in cell culture level. Moreover, both clinical studies were retracted (https://www.nytimes.com/2020/06/04/health/coronavirus-hydroxychloroquine.html?auth=login-google). Therefor, we omitted the corresponding sentence from the discussion.